# Improving localization-based approaches for breast cancer screening exam classification

**Thibault Févry**[1]**, Jason Phang**[1]**, Nan Wu** [1]**, S. Gene Kim**[2]**, Linda Moy**[2]**, Kyunghyun Cho**[1]**, Krzysztof J. Geras**[2,1]

[1] *Center for Data Science, New York University*

[2] *Department of Radiology, New York University School of Medicine*

## Abstract

We trained and evaluated a localization-based deep CNN for breast cancer screening exam classification on over 200,000 exams (over 1,000,000 images). Our model achieves an AUC of 0.919 in predicting malignancy in patients undergoing breast cancer screening, reducing the error rate of the baseline (Wu et al., 2019a) by 23%. In addition, the models generates bounding boxes for benign and malignant findings, providing interpretable predictions.

## 1. Introduction

Breast cancer is the second leading cause of cancer-related death among women in the United States. Screening mammography has been highly effective in reducing mortality rate but suffers from a high rate of false positive recalls, leading to added costs and stress. Better computer-aided diagnostic tools could improve patient outcomes by helping radiologists. Our goal is to build a system that provides highly accurate as well as interpretable predictions, in order to assist radiologists in cancer detection. To this end, we train an object detection network to predict the location of suspicious lesions and classify them.

## 2. Data

Our dataset (Wu et al., 2019b) comprises 229,426 screening mammography exams (1,001,093 images, with each exam containing the four standard views) from 141,473 unique patients. Among these, 5,832 exams had at least one biopsy performed within 120 days of a screening mammogram. Within this set of exams, 985 breasts had malignant findings, 5,556 had benign findings and 234 had both. For these exams, radiologists retrospectively annotated the locations of the biopsied lesions at a pixel level.

## 3. Methods

We used bounding boxes on the annotations of biopsied lesions to train a Feature-Pyramidal Network (Lin et al., 2017)-based Faster-RCNN model (Ren et al., 2015). In most experiments, we used a ResNet-50 (R-50; He et al., 2016) backbone pretrained on ImageNet (Deng et al., 2009) to initialize our models. We also conducted experiments with ResNet-101 (R-101) and ResNeXt-101 (X-101; Xie et al., 2017) backbones. The final classifier predicts three classes: benign lesions, malignant lesions, or nothing. We used `maskrcnn-benchmark` (Massa and Girshick, 2018) to train our models. We evaluated models after every 5,000

mini-batches and select the best models according to the score computed on the validation set. We used the standard setup of the framework, aside from the following exceptions.

**Use of non-annotated images** We used images without annotated lesions to train our model. These are treated as negatives for both the region-proposal network (RPN) and the classifier branch. The goal of this approach is to reduce overfitting on the comparatively small set of annotations. For our base experiment, during training, half of the images are sampled from an exam with a biopsy and half without.

**Resolution** We used a maximum resolution of $2200 \times 3000$ for the R-50 backbone, by resizing the mammograms with bilinear interpolation ($1700 \times 2700$ for R-101, $1300 \times 2100$ for X-101). Indeed, prior work (Geras et al., 2017) highlighted the importance of operating at a high resolution. Consequently, we also reduced the batch size to 4 (2 per GPU) due to our model's large memory requirements.

**IoU thresholds** We also relaxed the IoU threshold for foreground objects in the RPN from 0.7 to 0.5, as in (Ribli et al., 2018; Morrell et al., 2018). Compared to natural images, our data set has (i) noisier annotations with less precise boundaries, and (ii) fewer annotations per image. To circumvent (i), we also tried isotropic bounding box rescaling by a factor drawn uniformly at random in $[0.8, 1.2]$. We set the IoU threshold of the final non-maximum suppression to 0.1, following the rationale of Ribli et al. (2018) that for mammograms, "overlapping detections are expected to happen less often than in usual object detection".

**Optimization** We generally did not tune the learning rate schedules according to the advice of Goyal et al. (2017) ("recommended lr schedule") despite lowering the batch size. Indeed, found that it did not strongly impact results while considerably increasing training time. We used gradient clipping (for norm $> 3$) in all our experiments. We experimented with using batch normalization but found it not to help in our experiments.

**Inference** During inference, we take the maximum malignant prediction among the bounding boxes of each view and then average over both views to get the predicted probability for the whole breast. We found this setup to perform better than others, including taking the maximum over views or the mean over boxes. In addition, we found it useful to decrease the score threshold at inference time from 0.05 to 0.001, as the tail-like behavior is important when computing AUC.

## 4. Results

The results are shown in Table 1. Our methods compare favorably against the models of Wu et al. (2019a). Indeed, we achieve a 23% relative error reduction in AUC on the test set. Note that both models use pixel-level annotations. Due to high variance, we ensemble all runs together for our "Ensemble" setup. On the subpopulation of the test set used for the reader study in Wu et al. (2019a), our results also compare favorably to both earlier models and radiologists in terms of ROC, although the improvements are smaller. The reader study setup contains more exams from patients who underwent biopsy. Thus we believe this model distinguishes better between negative cases and cases that need a biopsy but is not as good in distinguishing between benign and malignant cases within the biopsied population. Ensembling our method with Wu et al. (2019a) improves the reader study AUC further to 0.895, compared to 0.778 for the average radiologist.

| model | test set | | reader study | |
|---|---|---|---|---|
| **Wu et al. (2019a) single model** | 0.886 | ± 0.003 | - | |
| **Wu et al. (2019a) ensemble** | 0.895 | | 0.876 | |
| **Base setup** | 0.891 | ± 0.005 | 0.845 | ± 0.007 |
| **+ biopsy ratio 0.75** | 0.895 | ± 0.004 | 0.855 | ± 0.012 |
| **+ biopsy ratio 1** | 0.887 | ± 0.011 | 0.855 | ± 0.013 |
| **+ bounding box scaling** | 0.890 | ± 0.006 | 0.841 | ± 0.010 |
| **+ classifier*5** | 0.903 | ± 0.007 | 0.859 | ± 0.013 |
| **+ bb scaling, classifier*5, ratio 0.75** | 0.888 | ± 0.006 | 0.855 | ± 0.006 |
| **+ recommended lr schedule** | 0.897 | ± 0.007 | 0.858 | ± 0.007 |
| **+ R-101 backbone** | 0.887 | ± 0.011 | 0.834 | ± 0.011 |
| **+ X-101 backbone** | **0.908** | ± 0.014 | **0.866** | ± 0.020 |
| **Ensemble** | **0.919** | | **0.879** | |
| **Ensemble + Wu et al. (2019a) ensemble** | **0.930** | | **0.895** | |

**Table 1:** AUC comparison against earlier results on the test set and reader study. Means and standard deviations for our models are computed over 3 random initializations. Base refers to the R-50 setup. For biopsy ratio, we increased the proportion of images from exams with biopsies in training. For classifier*5, we quintuple the weight on the classifier loss.

## 5. Analysis

Our validation metrics varied significantly between checkpoints. We believe that this is due to (i) the interaction of the components of the training loss, (ii) components of the training loss being only loosely related to the final metric, (iii) a small batch size rendering optimization unstable. This motivates ensembling all runs to produce our final model.

We notice a clear trade-off between using larger networks and using a larger resolution, with the R-50, R-101 and X-101 setups performing competitively. Unfortunately, due to computational constraints, we were unable to test R-101 and X-101 on the same resolution as R-50. Overall, our ensemble combining different backbones and resolutions performed best.

In Figure 1, we show an annotation and the corresponding prediction using an X-101 model. The model accurately predicts a ma-

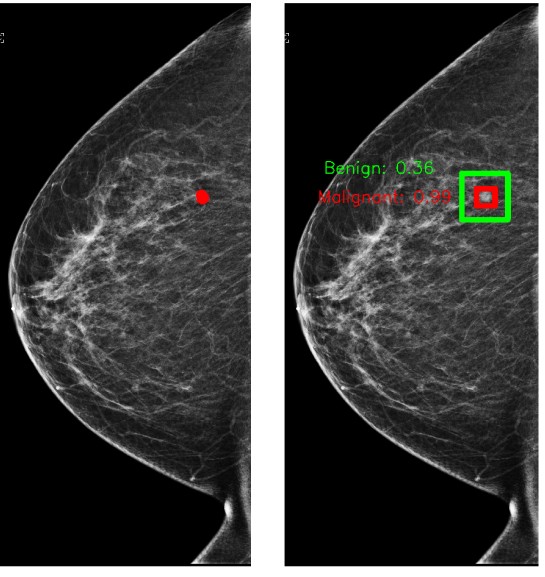

**Figure 1:** An example of an annotation of a malignant lesion (left) and predictions (right) from a X-101 model on a test set breast.

lignant lesion with high probability (0.99). It also predicts a benign lesion with low probability (0.36) for which there is no ground-truth annotation. These bounding boxes can highlight suspicious regions and help radiologists understand predictions from our models.

## Acknowledgments

We would like to thank Yiqiu Shen and Jungkyu Park for comments on earlier versions of this manuscript. We also gratefully acknowledge the support of Nvidia Corporation with the donation of some of the GPUs used in this research. This work was supported in part by grants from the National Institutes of Health (R21CA225175 and P41EB017183).

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
