# OpenReview forum: "Improving localization-based approaches for breast cancer screening exam classification"
_MIDL.io/2019/Conference/Abstract — MIDL Abstract 2019_

### Official Review · AnonReviewer1 · 2019-04-28
**improve originality (beyond reusing Lin CVPR'17), and evaluation of tumor localization on images (lacking) - perhaps still work in progress**

**Rating:** 2
**Confidence:** 3

**Review:**

Improving localization-based approaches for great cancer screening exam classification

The paper trains a feature-pyramidal network with variant resnet backbones, to learn the location of bounding boxes of benign and malign breast tumors in 2D mammograms. The method uses a recent architecture (Lin, cvpr'17) with various resnet architectures. Results are mitigated with little general improvement (AUC in ROC) from 0.89 to 0.91 or 0.87 to 0.87.
The submission may benefit from clarification on 1) originality of the method, 2) quantitative evaluation on the improvements of the tumor localization on the image (currently, lacking). As is, the paper may be perceived as incomplete to be convincing enough.

---

### Official Review · AnonReviewer2 · 2019-04-29
**Impressive dataset with good results, some important details missing**

**Rating:** 3
**Confidence:** 3

**Review:**

The authors present a method to classify mammograms and present bounding boxes of suspect lesions at the same time. The dataset size is impressive as are the results. An important criticism is that it unclear from the paper how data splits were done (e.g. training, validation, test). A bit to much knowledge needs to be gained from reading Wu et al.

---

### Decision · Program_Chairs · 2019-05-06
**Acceptance Decision**

Accept